# The Influence of Sodium Hexametaphosphate Chain Length on the Physicochemical Properties of High-Milk Protein Dispersions [note 1]

**DOI:** 10.3390/foods13091383

**Published:** 2024-04-30

**Authors:** Baheeja J. Zaitoun, Jayendra K. Amamcharla

**Affiliations:** 1Department of Animal Sciences and Industry, Food Science Institute, Kansas State University, Manhattan, KS 66506, USA; bzaitoun@ksu.edu; 2Midwest Dairy Foods Research Center, Department of Food Science and Nutrition, University of Minnesota, Saint Paul, MN 55108, USA

**Keywords:** milk protein concentrate, chelating agent, chain length

## Abstract

Protein–protein and protein–mineral interactions can result in defects, such as sedimentation and age gelation, during the storage of high-protein beverages. It is well known that age gelation can be delayed by adding cyclic polyphosphates such as sodium hexametaphosphate (SHMP). This study aims to assess the influence of different phosphate chain lengths of SHMP on the physicochemical properties of high-protein dispersions. The effect of adding different SHMP concentrations at 0%, 0.15%, and 0.25% (*w*/*w*) before and after heating of 6%, 8%, and 10% (*w*/*w*) milk protein concentrate dispersions was studied. The phosphate chain lengths of SHMPs used in this study were 16.47, 13.31, and 9.88, and they were classified as long-, medium-, and short-chain SHMPs, respectively. Apparent viscosity, particle size, heat coagulation time (HCT), color, and turbidity were evaluated. It was observed that the addition of SHMP (0.15% and 0.25%) increased the apparent viscosity of MPC dispersions. However, the chain length and the concentration of the added SHMP had no significant (*p* > 0.05) effect on the apparent viscosity after heating the dispersions. The HCT of a dispersion containing 6%, 8%, and 10% protein with no SHMP added was 15.28, 15.61, and 11.35 min, respectively. The addition of SHMP at both levels (0.15% and 0.25%) significantly increased the HCT. Protein dispersions (6%, 8%, and 10%) containing 0.25% short-chain SHMP had the highest HCT at 19.29, 19.61, and 16.09 min, respectively. Therefore, the chain length and concentration of added SHMP significantly affected the HCT of unheated protein dispersion (*p* < 0.05).

## 1. Introduction

The demand for high-protein beverages is increasing worldwide, driven by various factors such as convenience, the desire for a “healthier” lifestyle, population growth, and expanded urbanization [1]. Milk protein concentrate (MPC) has an 80:20 casein-to-whey protein ratio in milk, making it a preferable ingredient in formulating high-protein dairy beverages. In addition, MPC is heat stable compared with whey-derived ingredients. However, protein–protein and protein–mineral interactions that can occur because of using MPC as a protein source can result in defects during the storage, such as sedimentation and age gelation [2]. Sedimentation is usually explained by the formation of a protein-rich dense layer at the bottom of the beverage package due to the aggregation of k-casein-depleted casein micelles [3]. Sedimentation can be reversible upon mixing during the early stages of storage. The other storage defect that occurs in high-protein beverages is age gelation. Datta and Deeth (2001) [4] explained age gelation in UHT milk as a four-stage physical change. In the initial stage, some product thinning is observed for a short period, followed by a change in the viscosity in the second stage, which is usually an extended period. Subsequently, a sudden change in the viscosity with a gel formation occurs in the third stage, and in the final stage viscosity decreases as the gel matrix breaks down, leaving a serum layer and protein curds. There are two ways to improve the stability of high-dairy beverages: either by modifying the functional properties of the ingredients, such as enzymatic hydrolysis and chemical modification [5], or by adding some additives, such as emulsifying salts [6]. The Code of Federal Regulations (CFR; 21CFR133.169) classified emulsifying salts as monophosphates, condensed phosphates, glassy (long-chain) phosphates, citrates, and tartrates [7]. Those emulsifying salts’ ability to chelate calcium ions varies depending on the valence, chain length of ion species formed, pH, ionic strength, and temperature [8]. Cano-Ruiz and Richter (1998) [6] reported that they could delay age gelation by adding cyclic polyphosphates such as SHMP. Various studies have been performed on the use of SHMP in high-protein beverages. However, it was necessary to conduct a study that evaluates the influence of using different phosphate chain lengths of SHMP on the physicochemical properties of high-protein dispersions. This led us to our objective of the present study. We aimed to assess the effect of using small, medium, and long chains of SHMP on the physicochemical properties of high-protein dispersions.

## 2. Materials and Methods

### 2.1. Experimental Design

Three randomly selected lots of MPC 85 were obtained from a commercial manufacturer (Idaho Milk Products, Jerome, ID, USA). The manufacturer provided the MPC compositions, which contained about 86.77%, 4.97%, 1.12%, 4.77%, and 6.68% protein, moisture, fat, lactose, and ash content, respectively. Different SHMP powders were obtained from Prayon (Lyon, France). The phosphate chain lengths of SHMPs were 16.47, 13.31, and 9.88 and were classified as long, medium, and short SHMPs, respectively. The phosphate chain length was measured using phosphorus-31 nuclear magnetic resonance (^31^P-NMR) at the University of Kansas following the method explained by Stover et al. [9]. First, 1400 mL protein dispersions (6%, 8%, and 10% *w*/*w*) were prepared by dissolving the required amount of MPC 85 in distilled water at 45 °C for 30 min, then the aliquot was refrigerated (4 °C) overnight for complete rehydration. Samples were warmed up to 45 °C, and then the pH of the protein solution was adjusted to 7.00 ± 0.05 by adding 0.5 N of sodium hydroxide. Subsequently, the sample was divided equally into seven beakers containing 200 mL of the MPC 85 solution. Samples were placed on hot plates to maintain their temperature at 45 °C. As per the experimental design, the required quantity of SHMP (0%, 0.15%, and 0.25% *w*/*w* based on MPC solution) was weighed and added to the MPC solution under stirring for 10 min. The heat treatment was carried out in heat-resistant 8 mL glass vials (catalog#W224584; DWK Life Sciences, Millville, NJ, USA). MPC solution (3 mL) was poured into glass vials with a heat-resistant screw cap and immersed in an oil bath (Narang Scientific Works Pvt. Ltd., New Delhi, India) maintained at 140 °C. The actual temperature of the MPC solution in one of the vials was monitored using a thermocouple, and when the temperature reached 140 °C, a timer was set for 15 s. The come-up time was approximately 3 min. After 140 °C heat treatment for 15 s, the samples were immediately transferred to an ice bath to cool. All the experiments were performed in triplicate using the three lots of MPC 85. Samples were tested for apparent viscosity, heat coagulation time (HCT), mean particle size and zeta potential, FAST index, color, and turbidity.

#### 2.1.1. Heat Coagulation Time (HCT)

For this analysis, a 2 mL sample was transferred into a glass vial and immersed in an oil bath with a rocker (Narang Scientific Works Pvt. Ltd., New Delhi, India) at 140 °C. The time required for visual coagulation was recorded as HCT (min).

#### 2.1.2. Apparent Viscosity

The apparent viscosity of the heated and unheated samples was measured using a stress–strain-controlled rheometer (MCR-92 Anton Paar, Vernon Hills, IL, USA) with cone and plate geometry with 680 µL of the sample. The analysis was performed at 20 °C with a 0.1–200 s^−1^ shear rate. Apparent viscosity was reported at 100 s^−1^, and the analysis was performed in duplicate on each sample.

#### 2.1.3. Mean Particle Size and Zeta Potential

Dynamic light scattering (DelsaMax PRO, Beckman, Germany) with DelsaMax ASSIST was used in measuring the mean particle size and zeta potential for heated and unheated high-protein dispersions following the method described by Singh et al. [10]. Samples were diluted to 1:100 with distilled water and slowly injected into the flow cell using a syringe. The measurements were performed in triplicate.

#### 2.1.4. FAST Index

The FAST index assesses protein denaturation and Maillard reaction products in the heat-treated milk [11]. Tryptophan and Maillard products’ fluorescence spectra were attained using an LS50B luminescence spectrometer (PerkinElmer, Waltham, MA, USA). Then, the FAST index was calculated using the following Equation (1):(1)$$FAST\;index=\frac{Maillard\;products\;fluorescence}{Tryptophan\;fluorescence}$$

Front-Face Fluorescence Spectroscopy. The fluorescence spectra of tryptophan and Maillard products were acquired in the front-face geometrical configuration. In the front-face configuration, both the emission and excitation occur at the same cuvette face and thereby make it suitable for opaque samples such as MPC solutions. The instrument was equipped with a 15 W xenon lamp and the front-face accessory with a 56° incidence angle adjustment. The sample was loaded in a quartz cuvette (Catalog no: 3-Q-20; Starna Cells Inc., Atascadero, CA, USA) with a 20 mm path length. The fluorescence emission spectra of tryptophan were obtained at a wavelength of 290 nm for excitation and an emission range of 305–450 nm. The fluorescence emission spectra of Maillard products were obtained at a wavelength of 360 nm for excitation and an emission range of 380–480 nm. Seven scans were performed on each sample and averaged to improve the signal-to-noise ratio [10]. The excitation and emission slit widths were set at 10.0 and 7.0 nm, respectively. The 1% attenuation filter was selected, and the FL Data Manager Software (version 4.00.02, Perkin-Elmer) was used for the spectral data acquisition. Tryptophan fluorescence spectra of unheated and heated samples were obtained at a wavelength of 290 nm for excitation and at 340 nm and 350 nm emission wavelengths, respectively, and the fluorescence of advanced Maillard products measured at 360 for excitation and 430 nm for emission were used to calculate the FAST index using Equation (1).

Right-Angle Fluorescence Spectroscopy. Right-angle fluorescence spectroscopy implies measuring a transparent sample (serum phase) with an optical density of less than 0.2, so Beer–Lambert’s law can be applied to the quantitative assessment [10]. Therefore, samples were precipitated using 0.1M sodium acetate buffer at 4.6 pH, as explained by Singh et al. [10]. The instrument had a 15 W xenon lamp and a right-angle accessory. The sample was loaded in a quartz cuvette (Starna Cells Inc., Atascadero, CA, USA) with a 10 mm path length. The fluorescence emission spectra of tryptophan were obtained at a wavelength of 290 nm for excitation and an emission range of 305–450 nm; the excitation and emission slit widths were set at 3 nm and 3 nm, respectively. The fluorescence emission spectra of Maillard products were obtained at a wavelength of 360 nm for excitation and an emission range of 380–480 nm; the excitation and emission slit widths were set at 10 and 7 nm, respectively. Seven scans were performed on each sample and averaged to improve the signal-to-noise ratio. The open filter was used in the analysis. Tryptophan fluorescence of unheated and heated samples at 290/340 nm and 290/350 nm, respectively, and the fluorescence of advanced Maillard products measured at 360/430 nm were used to calculate the FAST index using Equation (1).

#### 2.1.5. Color (△E)

The CIE LAB values *L**, *a**, and *b** were measured using a Hunter-Lab Mini Scan colorimeter (Hunter Associates Laboratory, Reston, VA, USA). About 2 mL of the sample was placed in a transparent plastic container, and the measurement was taken. *L**, *a**, and *b** values were used in calculating the total change in the color (△E) using the following Equation (2):(2)$$△E=\sqrt{{\left(L_{1}-L_{2}\right)}^{2}+{\left(a_{1}-a_{2}\right)}^{2}+{\left(b_{1}-b_{2}\right)}^{2}}$$

#### 2.1.6. Turbidity

Turbidity was measured using a UV-5200 UV/Vis spectrophotometer (Metash Instruments Co., Ltd., Shanghai, China). Samples were diluted 1:10 with distilled water; then, the measurements were carried out at room temperature using a 700 nm wavelength and a plastic disposable cuvette (Fisher Scientific, Pittsburgh, PA, USA).

### 2.2. Statistical Analysis

All data were analyzed based on the completely randomized design (CRD) using PROC GLIMMIX in SAS Studio (version 9.4; SAS Inst., Cary, NC, USA) and Tukey’s test to determine any significant differences between treatment levels, which were declared significant when *p* ≤ 0.05.

## 3. Results

### 3.1. Heat Coagulation Time (HCT)

Heat stability is the most critical processing parameter, and it is considered the main factor in choosing the source of protein in high-protein dairy-based beverages. The HCT is the total time before the milk sample forms a visible clot after being placed in an oil bath maintained usually at 140 °C. Other methods can be used in determining heat stability, such as an ethanol test, a whitening test, protein sedimentation, and a viscosity determination [12]. After heating, mineral equilibrium between the serum and colloidal phase changes, besides the dissociation of k-casein from the casein micelle surface, which may cause a stearic destabilization of casein micelles [13].

The MPC dispersion containing 6% protein with no SHMP added had an HCT of 15.28 min (Table 1). It was observed that the addition of SHMP increased the HCT as the addition of calcium chelators reduces the concentration of free calcium ions and decreases the susceptibility to ultra-high temperatures during the processing [8]. Dispersions containing 0.15% of long, medium, and small SHMP had an HCT of 16.23, 17.26, and 17.69 min, respectively. The sample containing 0.15% long SHMP differed statistically from the medium and small dispersions. The HCT of MPC dispersions with 0.25% long SHMP was 17.13 min and significantly lower compared to MPC dispersions containing medium and short SHMP. Therefore, the chain length and concentration of added SHMP significantly affected (*p* < 0.05) the dispersions containing 6% protein.

The MPC dispersion containing 8% protein without SHMP had an HCT of 15.61 min. Dispersions containing 0.15% long, medium, and small SHMP had an HCT of 16.74, 17.27, and 18.62 min, respectively. The chain length of the added SHMP at the 0.15% level had no significant effect (*p* > 0.05) on the HCT. The HCT of 0.25% long, medium, and small dispersions was 17.23, 18.75, and 19.61 min, respectively. The sample containing 0.25% small SHMP had an HCT of 19.61, the highest. The chain length of the added SHMP at the 0.25% level had a significant effect (*p* < 0.05) on the HCT. Therefore, the chain length and concentration of the added SHMP have a significant impact (*p* < 0.05) on the HCT of 8% MPC dispersion.

The MPC dispersion containing 10% protein with no SHMP added had an HCT of 11.35 min. As the protein content increased, the HCT decreased due to increased calcium ion activity and heat-induced dissociation of the κ-casein [14]. The addition of the SHMP statistically increased the HCT. The range of HCT for dispersions containing 0.15% and 0.25% long, medium, and short SHMP was 14.28–16.09 min. The concentration of added SHMP significantly affected the HCT in 10% MPC dispersions. However, the chain length did not.

### 3.2. Apparent Viscosity

Table 2 shows the apparent viscosity of heated and unheated MPC dispersions at a 100 s^−1^ shear rate. Before heating, the MPC dispersion containing 6% protein with no SHMP had an apparent viscosity of 5.36 mPa·s. It was observed that the addition of different SHMP at both levels (0.15% and 0.25%) had statistically increased the apparent viscosity of MPC dispersions to 11.85–13.69 mPa·s in agreement with Pandalaneni et al. (2018) [15]. This increase in the apparent viscosity was related to the dissociation and swelling of casein micelles, which increases the effective volume fraction of the dispersed phase, leading to increased viscosity. The concentration of the SHMP had a significant (*p* < 0.05) effect on the apparent viscosity in unheated MPC dispersions containing 6% protein. However, the chain length did not.

Upon heating, the apparent viscosity of the MPC dispersion containing 6% protein with no SHMP was decreased to 2.49 mPa·s. It was statistically different from samples containing different SHMP at both levels and had a viscosity of 2.87–3.10 mPa·s. The decrease in the viscosity upon heating can be attributed to a higher degree of flexibility within the casein micelles and reduced interactions between casein micelles due to the reduction in intra- and intermolecular hydrophobic interactions, respectively [16]. These dispersions were significantly different from the sample with no SHMP added. However, the chain length of the added SHMP had no significant (*p* > 0.05) effect on the apparent viscosity in heated MPC dispersions containing 6% protein.

Before heating, the MPC dispersion containing 8% protein with no SHMP added had an apparent viscosity of 11.75 mPa·s, which was a little higher than 6.62 mPa·s as observed by Pandalaneni et al. (2018) [15]. When 0.15% and 0.25% long, medium, and small SHMP were added, the apparent viscosity increased, as observed in the 6% MPC dispersions. However, this increase was not significantly different (*p* > 0.05) from the dispersion with no SHMP added, and the chain length of added SHMP did not influence the dispersion’s apparent viscosity at 0.15%. Adding 0.25% resulted in a more significant increase in the apparent viscosity in unheated MPC dispersions. These dispersions differed significantly (*p* < 0.05) from the dispersion containing no SHMP. The chain length of added SHMP influenced the dispersion’s apparent viscosity when added at 0.25%. The SHMP tends to form calcium casein phosphate complexes when presented at higher concentrations in the presence of Ca, in addition to the cross-linking that the SHMP forms with casein proteins [17]. After heating, the apparent viscosity of the 8% MPC dispersions decreased due to the protein rearrangement during heating. 

The MPC dispersion contained no SHMP, which had a viscosity of 5.19 mPa·s and was not different from other SHMP dispersions. Therefore, the chain length and concentration of added SHMP had no significant (*p* > 0.05) effect on the apparent viscosity of the heated 8% MPC dispersions.

Before heating, the MPC dispersion containing 10% protein with no SHMP added had an apparent viscosity of 15.08 mPa·s. As observed in the 8% MPC dispersions, adding 0.15% of different SHMP insignificantly increased (*p* > 0.05) the apparent viscosity and chain length. However, adding 0.25% SHMP significantly increased the apparent viscosity of the dispersions. Moreover, due to the high protein content and adding SHMP at 0.25%, gelation was observed in all SHMP-treated samples because SHMP has a high affinity for binding with Ca. In addition, the SHMP can bind with the caseins’ positively charged amino acid residues [18,19], which results in cross-linking between caseins and forming the gel at a high concentration of casein and SHMP. The apparent viscosity after heating the 10% MPC dispersions had decreased due to the protein rearrangement during heating, similar to what was observed in 6% and 8% MPC dispersions. The MPC dispersion containing no SHMP had a viscosity of 4.68 mPa·s, which was not different from dispersions containing SHMP. Overall, the addition of SHMP significantly increased the viscosity in unheated dispersions. Heating the samples decreased the apparent viscosity of all samples. Moreover, adding different chain lengths of SHMP did not influence the viscosity in both heated and unheated dispersions.

### 3.3. Mean Particle Size

Table 3 shows the mean particle size of heated and unheated dispersions. Before heating, the MPC dispersion containing 6% protein with no SHMP added had a mean particle size of 200.58 µm. It was observed that the addition of SHMP (0.15% and 0.25%) decreased the particle size of MPC dispersions because the SHMP binds to calcium ions in the serum phase, which causes an imbalance in calcium ion concentration between the serum and colloidal phase resulting in the migration of calcium ions from the colloidal to the serum phase, which leads to the dissociation of casein micelles into submicelles and nonmicellar casein proteins [15]. A further decrease in the particle size was observed when a higher concentration of SHMP was added. Therefore, the added SHMP concentration significantly impacted the particle size of unheated MPC dispersions containing 6% protein. However, the chain length of added SHMP did not significantly affect the particle size (*p* > 0.05) at both levels.

Upon heating at 140 °C for 15 s, the sample with no SHMP added had a particle size of 158.58 µm. The particle size of SHMP-treated samples is in the range of 123.22–146.97 µm. The added SHMP concentration significantly affected the particle size of heated MPC dispersions containing 6% protein. However, the chain length of added SHMP did not significantly affect the particle size (*p* > 0.05).

Before heating the MPC dispersion containing 8% protein and no SHMP added, the particle size was 193.88 µm and was not statistically different from dispersions containing 0.15% SHMP. However, adding 0.25% SHMP decreases the particle size to 105.42–140.53 µm. Therefore, the concentration of added SHMP statistically decreased the particle size; however, the chain length of added SHMP did not (*p* > 0.05). When those samples were heated at 140 °C for 15 s, the particle size of samples containing 0.15% long, medium, and small SHMP was 136.75–146.62 µm. When 0.25% long, medium, and small SHMPs were added, the particle size was 169.23–177.65 µm. The chain length of added SHMP at both levels did not statistically (*p* > 0.05) affect the particle size of heated dispersions containing 8% protein.

Before heating, the MPC dispersion containing 10% protein with no SHMP added had a particle size of 199.52 µm, similar to the dispersions containing 0.15% SHMP. Similar to 6% and 8% MPC dispersions, the addition of SHMP decreased the particle size. Adding more SHMP (0.25%) caused a further significant decrease in particle size. It ranged from 100.37–102.55 µm when different chain lengths of SHMP were added. The chain length of added SHMP did not affect the particle size of unheated MPC dispersions containing 10% protein at both levels. When those dispersions were heated, the sample with no SHMP added had a particle size of 172.33 µm, similar to the dispersions containing 0.15% SHMP, and the particle size was 170.60–178.05 µm. Meanwhile, samples containing 0.25% SHMP had particle sizes of 213.12–225.22 µm. The chain length of added SHMP at both levels did not statistically (*p* > 0.05) affect the particle size of heated dispersions containing 10% protein.

### 3.4. Zeta Potential

The zeta potential of both heated and unheated dispersions is shown in Table 4. Unheated 6% protein dispersion containing no SHMP had a zeta potential of −18.43 mV. When the SHMP was added, the zeta potential significantly increased from −21.22 to −24.90 mV. The concentration of added SHMP influenced the zeta potential in unheated 6% protein dispersion with no effect on the phosphate chain length of SHMP observed. Upon heating, the same trend was observed. The sample containing no SHMP had a zeta potential of −18.96 mV. Zeta potential increased significantly in samples containing the SHMP, ranging from −24.56 to −25.55 mV. Therefore, the concentration of added SHMP influenced the zeta potential in heated 6% protein dispersion, but the chain length of added SHMP did not. Choi and Zhong (2020) [20] reported that when SHMP was added to 5% skim milk powder dispersions at higher concentrations, there was an increase in the phosphate and calcium in the serum phase, which increased the ionic strength that lowered electrophoretic mobility and, therefore, the measured zeta potential for those dispersions.

Unheated 8% protein dispersion containing no SHMP had a zeta potential of −20.25 mV. When the SHMP was added, the zeta potential was in the −22.78 to −25.92 mV range. However, this increase was insignificant. Therefore, neither the concentration nor the chain length of added SHMP influenced the zeta potential in unheated 8% protein dispersion. When those dispersions were heated, the zeta potential of the sample containing no SHMP was −20.99 mV. When 0.15% of different SHMP was added, there was an insignificant increase in the zeta potential. However, when 0.25% was added, a significant rise in the zeta potential was observed, ranging from −25.07 to −26.04 mV. Similar to 6% protein dispersions, the concentration of added SHMP influenced the zeta potential in heated 8% protein dispersion, but the chain length did not.

Unheated 10% protein dispersion containing no SHMP had a zeta potential of −18.43 mV. When 0.15% SHMP was added, the zeta potential significantly increased, and it was in the range of −21.22 to −23.20 mV. A further significant increase in zeta potential was observed when 0.25% SHMP was added. Upon heating, the zeta potential of the sample containing no SHMP was −20.62 mV. When 0.15% and 0.25% SHMP were added, a significant increase was observed in all samples except samples containing 0.15% medium and small SHMP with −23.84 and −23.10, respectively. Therefore, the concentration of added SHMP influenced the zeta potential in unheated and heated 10% protein dispersion, but the phosphate chain length did not.

### 3.5. FAST Index

The FAST index is a quick method that quantifies protein denaturation using the fluorescence of advanced Maillard reaction products and soluble tryptophan (FAST) [21].

#### 3.5.1. Front-Face Fluorescence Spectroscopy

The FAST index of heated and unheated samples using the front face is shown in Table 5. In dispersion containing 6% protein, the FAST index of the sample containing no SHMP was 4.36. Adding 0.15% SHMP showed no significant change in the FAST index. However, adding 0.25% SHMP significantly decreased the FAST index, which was in the 3.81–3.94 range. Therefore, the concentration of added SHMP has a significant effect (*p* < 0.05) on the FAST index of unheated dispersions containing 6% protein. However, the chain length of SHMP showed no impact. The increase in heating time and temperature intensifies the fluorescence of the thermally treated products [22]. Therefore, the FAST index increased in all samples upon heating. For example, the FAST index of the sample containing no SHMP increased from 4.36 to 9.35. The same increase was observed in samples containing different chain lengths of SHMP; the FAST index ranged from 8.84 to 9.56. Therefore, neither the concentration nor the chain length of added SHMP influenced the FAST index of the dispersions containing 6% when heated.

In 8% protein dispersions, the sample containing no SHMP had a FAST index of 4.76 and it was similar to all samples containing SHMP except for the sample containing 0.15% small-chain SHMP with a FAST index of 5.42. Therefore, the concentration of added SHMP significantly affected the FAST index of unheated 8% protein dispersions, but the chain length of added SHMP did not influence the FAST index. Upon heating, the FAST index insignificantly increased to 9.70–10.27. Therefore, the concentration and chain length of added SHMP had no significant effect (*p* > 0.05) on the FAST index of dispersions containing 8% protein. In 10% protein dispersions, the sample with no SHMP added had a FAST index of 5.09, with no significant effect observed when long, medium, and small SHMP were added at both levels (0.15% and 0.25%), as their FAST index was in the range of 5.56–7.49. Upon heating, the FAST index increased from 5.09 to 9.09 in the sample containing no SHMP. Samples containing long, medium, and small SHMP had a FAST index of 10.30–10.91. The chain length of added SHMP had no significant effect on the FAST index; however, the concentration did (*p* < 0.05).

#### 3.5.2. Right-Angle Fluorescence Spectroscopy

The FAST index of all the heated and unheated soluble phase protein samples is shown in Table 6. In the unheated sample containing 6% protein and no SHMP added, the FAST index was 23.71. The addition of different SHMP at different concentrations did not influence the unheated dispersions containing 6% protein, as the FAST index was 24.19–25.38. Upon heating, the FAST index of the sample containing no SHMP increased from 23.71 to 78.04. The FAST index increased due to the protein denaturation and Maillard products, which led to an increase in advanced Maillard products such as pyrrole and imidazole derivatives [23]. In addition, the tryptophan signal decreases as the protein is denatured. Samples containing SHMP had a FAST index in the 82.62–85.69, and they were all similar (*p* > 0.05) to the sample containing no SHMP except for the sample containing 0.25% long SHMP. Therefore, the added SHMP concentration significantly influenced the FAST index of heated 6% protein dispersions. However, the chain length did not.

In the unheated sample containing 8% protein and no SHMP added, the FAST index was 27.81, and neither the SHMP phosphate chain length nor the concentration added had any influence (*p* > 0.05) on the unheated dispersions containing 8% protein.

Upon heating, the FAST index of the sample containing no SHMP increased from 27.81 to 85.94. Samples containing SHMP had a FAST index of 89.16–98.95, and they were all similar (*p* > 0.05) to the sample containing no SHMP, except the sample containing 0.25% long and small SHMP. Therefore, the added SHMP concentration significantly influenced the FAST index of heated 8% protein dispersions. However, the chain length did not. In the unheated sample containing 10% protein and no SHMP added, the FAST index was 32.66. When SHMP was added to the protein dispersions, the FAST index was 28.67–33.65. Similar to unheated samples containing 6 and 8%, the addition of the SHMP did not influence the unheated dispersions containing 10% protein. In addition, heating increased the FAST index in those dispersions to 94.07–108.75. However, no significant influence was observed for the concentration or the chain length of added SHMP.

### 3.6. Color (△E)

The change in the color (△E) of MPC dispersions before and after adding different SHMP chain lengths is shown in Table 7. After adding 0.15% long, medium, and small SHMP, the color of the MPC dispersions containing 6% changed from 22.18 to 26.09. A further significant increase in △E value was observed when 0.25% of different chain lengths of SHMP were added; the △E was 43.01–43.89. The concentration had significantly changed the color of those dispersions. However, the chain length of SHMP did not affect the color change. The color change is due to the dissociation of the casein micelles, which results in a more translucent dispersion, as adding more SHMP results in binding more calcium ions in the serum phase, leading to more dissociation of the casein micelles. Upon heating, the color change decreased in all dispersions as they became milk-like dispersions. Samples containing 0.15% of different SHMP had a color change of 5.00–6.12. At the same time, dispersions containing 0.25% SHMP had a color change of 1.69–4.22.

Unheated MPC dispersions containing 8% and 10% protein behaved like dispersions containing 6%. The concentration had significantly changed the color of those dispersions. However, the chain length of SHMP did not affect the color change. When MPC dispersions containing 8% were heated, the ones containing 0.15% of different SHMP were similar to the sample containing no SHMP. However, adding 0.25% SHMP significantly increased △E compared to the sample containing no SHMP. Therefore, the concentration significantly changed the color of those dispersions. However, the chain length of SHMP did not affect the color change. When MPC dispersions containing 10% were heated, the ones containing 0.15% of different SHMP had a color change of 2.06–2.31. It differed statistically from the dispersions containing 0.25% SHMP, with a color change of 3.63–4.73. Therefore, the concentration significantly changed the color of those dispersions. However, the chain length of SHMP did not affect the color change.

### 3.7. Turbidity

Table 8 shows the change in the turbidity of unheated and heated protein dispersions before and after adding SHMP with different chain lengths. The change in the turbidity indicates the ability of emulsifying salts to chelate the calcium ions in protein dispersion.

This ability highly varies with the chain lengths of emulsifying salts. Mizuno and Lucey (2005) [24] noted that the SHMP had the highest ability to disperse casein. The turbidity of unheated MPC dispersions containing 6% protein with no SHMP added was 1.32. The addition of SHMP resulted in translucent dispersions that caused a significant decrease in turbidity. The turbidity was 0.36–0.40 when 0.15% long, medium, and small SHMP was added. A further significant decrease in turbidity was observed when 0.25% SHMP was added. Therefore, the chain length of added SHMP had no significant effect (*p* > 0.05) on the turbidity of unheated MPC containing 6% protein when both concentrations of SHMP were added. The decrease in the turbidity measurement is a sign of the casein dissociation caused by the removal of CCP from casein micelles, which reduced the refractive index of casein micelles [19]. While heating, casein aggregates and rearranges, which increases the turbidity measurement and causes the dispersion to be milk-like. The sample containing no SHMP had a turbidity of 1.40 and was significantly different from the turbidity of dispersions containing SHMP, which was 0.53–0.69. The turbidity of the sample containing 8% protein with no SHMP added was 1.56. Similar to our findings in 6% MPC dispersions, adding SHMP significantly decreased the turbidity. And the concentration of the SHMP had a significant effect (*p* < 0.05) on the turbidity measurement. However, the chain length of SHMP did not affect the turbidity measurement. When the MPC dispersions containing 8% protein were heated, the sample with no SHMP had a turbidity of 1.46. The turbidity of all SHMP-treated samples was similar to the sample with no SHMP except for the sample containing 0.15% small SHMP. Therefore, the chain length did not affect the turbidity of heated MPC dispersions containing 8% protein, but the concentration did. When the 10% protein samples were heated, the sample with no SHMP had a turbidity of 2.61, similar to samples with 0.25% SHMP, whose turbidity ranged from 2.76 to 2.87. Samples containing 0.15% long, medium, and small SHMP had turbidity in the range of 1.97–2.03, which was statistically different from the sample with no SHMP added. The chain length did not affect the turbidity of heated MPC dispersions containing 10% protein, but the concentration did.

## 4. Conclusions

In unheated protein dispersions, the apparent viscosity, turbidity, and △E significantly increased with the increased concentrations of added SHMP. At the same time, the particle size significantly decreased. When protein dispersions were heated, the concentration of added SHMP significantly influenced those properties. Dispersions containing 6 and 8% protein showed a significant increase in the HCT when SHMP with different chain lengths was applied. However, dispersions containing 10% protein were primarily affected by the concentration of the added SHMP rather than the chain length.

This might be due to the high concentration of proteins in that dispersion, as more SHMP might have been added to detect the influence of the phosphate chain length on its HCT. In conclusion, the phosphate chain length of SHMP mainly affected the stability of high-protein milk dispersions as it showed a higher HCT when the short phosphate chain length of SHMP was used. Therefore, using the short phosphate chain SHMP might be the most suitable emulsifying salt in products requiring high heat treatment, such as UHT and retort sterilized high-protein beverages.

## Figures and Tables

**Table 1 foods-13-01383-t001:** Heat coagulation time (min) of milk protein dispersions at 6%, 8%, and 10% protein levels.

SHMP Chain Length	SHMP Level (%)	Protein Content (%, *w*/*w*)		
		6	8	10
Control	0	15.28 ± 0.15 ^e^	15.61 ± 0.83 ^c^	11.35 ± 0.65 ^b^
Long	0.15	16.23 ± 0.48 ^d^	16.74 ± 1.58 ^bc^	14.28 ± 0.87 ^a^
	0.25	17.13 ± 0.70 ^cd^	17.23 ± 1.47 ^bc^	15.24 ± 1.12 ^a^
Medium	0.15	17.26 ± 0.47 ^c^	17.27 ± 1.35 ^abc^	14.98 ± 1.39 ^a^
	0.25	18.46 ± 0.98 ^ab^	18.75 ± 0.82 ^ab^	16.04 ± 0.23 ^a^
Short	0.15	17.69 ± 0.29 ^bc^	18.62 ± 1.24 ^ab^	15.38 ± 0.77 ^a^
	0.25	19.29 ± 0.86 ^a^	19.61 ± 1.09 ^a^	16.09 ± 0.26 ^a^

The values are presented as (mean ± SD). *n* = 3. ^a–e^ Means with different superscripts in the same column are significantly different (*p* < 0.05).

**Table 2 foods-13-01383-t002:** Apparent viscosity (mPa·s) of unheated and heated protein dispersions at 100 s^−1^ shear rate.

SHMP	Level (%)	6% Protein		8% Protein		10% Protein	
		Unheated	Heated	Unheated	Heated *	Unheated	Heated
Control	0	5.36 ± 2.26 ^b^	2.49 ± 0.09 ^b^	11.75 ± 4.63 ^c^	5.19 ± 2.25	15.08 ± 8.83 ^b^	4.68 ± 0.92 ^b^
Long	0.15	13.60 ± 4.04 ^a^	2.87 ± 0.09 ^ab^	27.04 ± 9.38 ^bc^	4.62 ± 1.21	39.62 ± 18.15 ^b^	7.08 ± 1.37 ^ab^
	0.25	12.30 ± 3.27 ^a^	3.10 ± 0.29 ^a^	40.56 ± 3.27 ^b^	4.53 ± 0.32	107.14 ± 50.15 ^a^	6.80 ± 1.63 ^ab^
Medium	0.15	11.96 ± 3.34 ^a^	2.99 ± 0.03 ^a^	25.14 ± 5.90 ^bc^	4.65 ± 0.98	38.10 ± 18.67 ^b^	6.77 ± 1.24 ^ab^
	0.25	13.69 ± 5.54 ^a^	3.02 ± 0.10 ^a^	63.41 ± 9.48 ^a^	5.36 ± 0.87	103.17 ± 52.07 ^a^	8.27 ± 2.44 ^a^
Short	0.15	12.33 ± 4.31 ^a^	2.93 ± 0.07 ^a^	28.99 ± 7.28 ^bc^	5.02 ± 0.56	41.70 ± 35.11 ^b^	6.73 ± 1.60 ^ab^
	0.25	11.85 ± 3.22 ^a^	3.08 ± 0.13 ^a^	40.12 ± 7.47 ^b^	5.77 ± 1.01	110.67 ± 61.93 ^a^	7.19 ± 0.51 ^ab^

The values are presented as (mean ± SD). *n* = 3. ^a–c^ Means with different superscripts in the same column are significantly different (*p* < 0.05). * No significant difference was observed within the column.

**Table 3 foods-13-01383-t003:** Mean particle size (µm) of unheated and heated protein dispersions at three different protein levels.

SHMP	Level (%)	6% Protein		8% Protein		10% Protein	
		Unheated	Heated	Unheated	Heated	Unheated	Heated
Control	0	200.58 ± 38.61 ^a^	158.58 ± 19.08 ^a^	193.88 ± 9.08 ^a^	150.23 ± 0.80 ^ab^	199.52 ± 48.49 ^a^	172.33 ± 18.72 ^cd^
Long	0.15	162.98 ± 18.29 ^b^	126.25 ± 2.15 ^c^	183.82 ± 16.11 ^ab^	146.62 ± 5.15 ^ab^	177.07 ± 9.72 ^a^	178.05 ± 14.37 ^bcd^
	0.25	102.47 ± 17.78 ^c^	137.77 ± 6.94 ^bc^	140.53 ± 49.43 ^bc^	176.95 ± 15.51 ^a^	101.87 ± 3.23 ^b^	214.63 ± 41.25 ^ab^
Medium	0.15	170.98 ± 22.05 ^ab^	123.22 ± 6.90 ^c^	183.58 ± 12.14 ^ab^	145.50 ± 6.17 ^ab^	172.12 ± 7.50 ^a^	170.60 ± 19.26 ^d^
	0.25	94.70 ± 10.24 ^c^	140.28 ± 6.96 ^abc^	118.43 ± 18.67 ^c^	169.23 ± 12.91 ^ab^	102.55 ± 8.08 ^b^	213.12 ± 38.88 ^abc^
Short	0.15	151.63 ± 29.17 ^b^	124.42 ± 4.08 ^c^	170.93 ± 9.69 ^ab^	136.75 ± 11.09 ^b^	173.03 ± 13.25 ^a^	172.43 ± 13.33 ^cd^
	0.25	103.55 ± 29.59 ^c^	146.97 ± 0.98 ^ab^	105.42 ± 8.25 ^c^	177.65 ± 18.36 ^a^	100.37 ± 8.90 ^b^	225.22 ± 42.15 ^a^

The values are presented as (mean ± SD). *n* = 3. ^a–d^ Means with different superscripts in the same column are significantly different (*p* < 0.05).

**Table 4 foods-13-01383-t004:** Zeta potential of unheated and heated protein dispersions.

SHMP	Level (%)	6% Protein		8% Protein		10% Protein	
		Unheated	Heated	Unheated *	Heated	Unheated	Heated
Control	0	−18.43 ± 0.75 ^a^	−18.96 ± 0.89 ^a^	−20.25 ± 2.12	−20.99 ± 1.05 ^a^	−18.43 ± 0.75 ^a^	−20.62 ± 1.01 ^a^
Long	0.15	−23.20 ± 1.21 ^b^	−25.33 ± 2.49 ^b^	−22.78 ± 0.43	−24.95 ± 2.16 ^ab^	−23.20 ± 1.21 ^b^	−25.91 ± 2.70 ^b^
	0.25	−24.90 ± 1.42 ^b^	−25.27 ± 1.33 ^b^	−25.92 ± 1.17	−25.73 ± 1.24 ^b^	−24.90 ± 1.42 ^d^	−25.43 ± 0.45 ^b^
Medium	0.15	−21.52 ± 0.76 ^ab^	−25.08 ± 2.36 ^b^	−25.88 ± 5.50	−24.75 ± 2.76 ^ab^	−21.52 ± 0.76 ^bc^	−23.84 ± 1.80 ^ab^
	0.25	−24.33 ± 0.73 ^b^	−25.55 ± 1.93 ^b^	−25.86 ± 0.73	−26.04 ± 1.96 ^b^	−24.33 ± 0.73 ^cd^	−24.23 ± 3.08 ^b^
Short	0.15	−21.22 ± 1.40 ^ab^	−24.94 ± 1.30 ^b^	−24.05 ± 0.61	−25.34 ± 2.37 ^ab^	−21.22 ± 1.40 ^b^	−23.10 ± 1.97 ^ab^
	0.25	−24.11 ± 4.05 ^b^	−24.56 ± 0.57 ^b^	−24.81 ± 2.16	−25.07 ± 1.61 ^ab^	−24.11 ± 4.05 ^d^	−24.69 ± 1.09 ^b^

The values are presented as (mean ± SD). *n* = 3. ^a–d^ Means with different superscripts in the same column are significantly different (*p* < 0.05). * No significant difference was observed within the column.

**Table 5 foods-13-01383-t005:** FAST index of unheated and heated protein dispersions using front-face fluorescence spectroscopy.

SHMP	Level (%)	6% Protein		8% Protein		10% Protein	
		Unheated	Heated *	Unheated	Heated *	Unheated *	Heated
Control	0	4.36 ± 0.94 ^ab^	9.35 ± 1.98	4.76 ± 1.10 ^b^	10.13 ± 1.73	5.09 ± 0.76	9.09 ± 1.15 ^b^
Long	0.15	4.76 ± 1.45 ^a^	9.32 ± 1.06	5.22 ± 1.21 ^ab^	10.27 ± 1.70	5.64 ± 1.45	10.75 ± 1.72 ^a^
	0.25	3.82 ± 1.20 ^c^	8.84 ± 1.00	4.96 ± 1.60 ^ab^	10.17 ± 2.56	5.71 ± 1.02	10.34 ± 1.74 ^ab^
Medium	0.15	4.62 ± 1.21 ^a^	9.02 ± 1.31	5.23 ± 1.15 ^ab^	9.89 ± 2.06	7.49 ± 2.60	10.30 ± 2.26 ^ab^
	0.25	3.81 ± 1.16 ^c^	9.26 ± 1.49	4.98 ± 1.47 ^ab^	10.02 ± 2.24	6.40 ± 1.28	10.43 ± 1.74 ^ab^
Short	0.15	4.58 ± 1.30 ^a^	9.33 ± 1.26	5.42 ± 1.35 ^a^	9.70 ± 1.22	5.71 ± 1.57	10.91 ± 1.90 ^a^
	0.25	3.94 ± 1.24 ^bc^	9.56 ± 1.34	4.82 ± 1.45 ^ab^	10.19 ± 2.40	5.56 ± 1.07	10.90 ± 1.97 ^a^

The values are presented as (mean ± SD). *n* = 3. ^a–c^ Means with different superscripts in the same column are significantly different (*p* < 0.05). * No significant difference was observed within a column.

**Table 6 foods-13-01383-t006:** FAST index of unheated and heated protein dispersions using right-angle fluorescence spectroscopy.

SHMP	Level (%)	6% Protein		8% Protein		10% Protein	
		Unheated *	Heated	Unheated *	Heated	Unheated *	Heated *
Control	0	23.71 ± 4.46	78.04 ± 12.43 ^b^	27.81 ± 5.54	85.94 ± 10.39 ^b^	32.66 ± 1.95	94.07 ± 14.28
Long	0.15	24.26 ± 4.19	83.54 ± 14.61 ^ab^	28.89 ± 6.37	89.16 ± 13.50 ^ab^	31.72 ± 2.76	99.28 ± 21.50
	0.25	25.01 ± 4.22	85.69 ± 16.81 ^a^	28.38 ± 4.94	98.95 ± 15.22 ^a^	29.98 ± 2.08	106.38 ± 24.04
Medium	0.15	24.19 ± 4.49	82.62 ± 14.34 ^ab^	29.77 ± 5.46	93.77 ± 12.32 ^ab^	33.13 ± 3.89	102.61 ± 14.65
	0.25	25.34 ± 4.18	84.67 ± 13.79 ^ab^	28.16 ± 5.31	96.93 ± 14.78 ^ab^	28.67 ± 2.19	108.75 ± 18.54
Short	0.15	24.78 ± 3.77	82.95 ± 16.26 ^ab^	28.42 ± 5.36	94.78 ± 15.27 ^ab^	33.65 ± 3.80	98.47 ± 19.98
	0.25	25.38 ± 3.84	84.72 ± 14.14 ^ab^	27.37 ± 4.86	98.77 ± 12.79 ^a^	31.65 ± 1.78	107.22 ± 23.32

The values are presented as (mean ± SD). *n* = 3. ^a,b^ Means with different superscripts in the same column are significantly different (*p* < 0.05). * No significant difference was observed within a column.

**Table 7 foods-13-01383-t007:** The color (△E) change of unheated and heated protein dispersions.

SHMP	Level (%)	6% Protein		8% Protein		10% Protein	
		Unheated	Heated	Unheated	Heated	Unheated	Heated
Control	0	0.00 ^c^	0.00 ^d^	0.00 ^c^	0.00 ^c^	0.00 ^c^	0.00 ^c^
Long	0.15	22.18 ± 2.47 ^b^	5.22 ± 1.75 ^a^	12.04 ± 3.73 ^b^	1.11 ± 0.28 ^bc^	10.20 ± 5.90 ^bc^	2.06 ± 0.83 ^b^
	0.25	43.89 ± 2.64 ^a^	4.22 ± 1.59 ^ab^	38.58 ± 1.45 ^a^	3.62 ± 1.11 ^a^	41.450 ± 6.22 ^a^	3.99 ± 0.27 ^a^
Medium	0.15	23.11 ± 2.55 ^b^	6.12 ± 1.18 ^a^	13.74 ± 4.36 ^b^	1.18 ± 0.47 ^bc^	11.22 ± 5.79 ^b^	2.18 ± 0.74 ^b^
	0.25	43.01 ± 3.65 ^a^	2.50 ± 0.61 ^bc^	37.58 ± 0.48 ^a^	2.98 ± 1.39 ^ab^	37.68 ± 6.73 ^a^	3.63 ± 1.02 ^a^
Short	0.15	26.09 ± 3.45 ^b^	5.00 ± 0.62 ^a^	17.38 ± 2.83 ^b^	1.69 ± 0.72 ^abc^	8.85 ± 7.69 ^bc^	2.31 ± 0.89 ^b^
	0.25	43.70 ± 2.61 ^a^	1.69 ± 0.69 ^cd^	39.52 ± 1.19 ^a^	3.31 ± 1.80 ^ab^	38.82 ± 5.66 ^a^	4.73 ± 0.63 ^a^

The values are presented as (mean ± SD). *n* = 3. ^a–d^ Means with different superscripts in the same column are significantly different (*p* < 0.05).

**Table 8 foods-13-01383-t008:** The turbidity of unheated and heated protein dispersions.

SHMP	Level (%)	6% Protein		8% Protein		10% Protein	
		Unheated	Heated	Unheated	Heated	Unheated	Heated
Control	0	1.32 ± 0.28 ^a^	1.40 ± 0.36 ^a^	1.56 ± 0.30 ^a^	1.46 ± 0.18 ^a^	2.16 ± 0.41 ^a^	2.61 ± 0.56 ^a^
Long	0.15	0.40 ± 0.16 ^b^	0.58 ± 0.07 ^b^	0.69 ± 0.23 ^b^	1.08 ± 0.21 ^ab^	0.96 ± 0.11 ^b^	2.02 ± 0.28 ^b^
	0.25	0.16 ± 0.09 ^c^	0.59 ± 0.07 ^b^	0.24 ± 0.12 ^c^	1.45 ± 0.27 ^a^	0.28 ± 0.09 ^c^	2.78 ± 0.33 ^a^
Medium	0.15	0.38 ± 0.15 ^b^	0.55 ± 0.09 ^b^	0.63 ± 0.23 ^b^	1.00 ± 0.21 ^ab^	1.01 ± 0.12 ^b^	1.97 ± 0.25 ^b^
	0.25	0.17 ± 0.10 ^c^	0.63 ± 0.09 ^b^	0.21 ± 0.09 ^c^	1.24 ± 0.32 ^ab^	0.29 ± 0.08 ^c^	2.76 ± 0.21 ^a^
Short	0.15	0.36 ± 0.17 ^b^	0.53 ± 0.07 ^b^	0.48 ± 0.12 ^bc^	0.86 ± 0.12 ^b^	1.04 ± 0.18 ^b^	2.03 ± 0.13 ^b^
	0.25	0.17 ± 0.09 ^c^	0.69 ± 0.04 ^b^	0.19 ± 0.09 ^c^	1.32 ± 0.32 ^ab^	0.30 ± 0.09 ^c^	2.87 ± 0.19 ^a^

The values are presented as (mean ± SD). *n* = 3. ^a–c^ Means with different superscripts in the same column are significantly different (*p* < 0.05).

## Data Availability

The original contributions presented in the study are included in the article, further inquiries can be directed to the corresponding author.

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
