# Peer review of "The Influence of Sodium Hexametaphosphate Chain Length on the Physicochemical Properties of High-Milk Protein Dispersions"

_foods, 2024, doi:10.3390/foods13091383_

Round 1

Reviewer 1 Report

Comments and Suggestions for Authors

This manuscript investigated the influence of sodium hexametaphosphate (SHMP) chain length on the physiochemical properties of high-milk protein dispersions. The experimental design performed is relatively simple and does not involve great technical difficulty. In general, the paper lacks scientific depth. There are many weaknesses. Furthermore, some of the results in the manuscript simply describe the experimental results, the reasons for the experimental results are not analyzed with sufficient references, such as sections 3.3, 3.5. Meanwhile, the results and discussions were almost exclusively in terms of the concentrations of the three protein dispersions (6%, 8%, and 10%), the respective analyses are verbose with many repetitive sentences and vague discussions, the sufficient discussion should be elaborated and refined. The following problems should be clarified.

1.       Line 15: In the abstract, the full name of "MPC 85" should be given when the first time it is used. And "Three fresh lots of MPC 85 were obtained from a commercial manufacturer.", why is the detail of MPC included in the abstract?

2.       Line 16: Change "SHMP at 0, 0.15, and 0.25% (w/w)" to "SHMP at 0%, 0.15%, and 0.25% (w/w)", and the same goes for line 17, line 23, etc.

3.       Be sparing with abbreviations in keywords and paper title: only abbreviations firmly established in the field may be eligible.

4.       Line 22: Change P>0.05 to P > 0.05. P should be italicized in the manuscript, and the same goes for line 28, line 175, etc.

5.       Line 29: I think the keyword "Beverage" is too broad, maybe you should use "high-protein beverage" or other specific word.

6.       All the temperatures are written without spaces behind the numbers, such as line 68, 70, 73, etc. A similar problem occurs in line 70 "7.00±0.05"so the author is asked to check the whole manuscript carefully.

7.       Line 76: Is this manufacturer's information for oil bath equipment?

8.       Line 92: Please add instrument parameters and sampling details in "2.1.3".

9.       Line 120: "as explained by [10]", please check the reference style and follow the guidelines of this journal.

10.     Line 134: "L*, a*, and b*", "L, a and b" should be in italicized in the manuscript.

11.     Line 154: What is the detail of P-NMR? And why are the details of phosphate and MPC included in this section? Is it more appropriate to put them in "2. Materials and Methods"?

12.     Line 168: Here it says "calcium ions", but in line 54 it says"Ca+2", so please write be consistent.

13.    Line 169-176: Why not describe the two SHMP levels (0.15 and 0.25%) together, since the trends are the same? And throughout the whole tables, the absence of spaces before and after the "±".

14.    In "3.1. Heat Coagulation Time (HCT)", is the reason for the increase in HCT caused by the SHMP chain length consistent at 6% and 8% protein content?

15.    Line 200: Change "rates" to "rate".

16.    Table 2: What is the meaning of the "*" after the 8% "Heated", which is not stated in the Note?

17.    In "3.2 Apparent Viscosity", the enumeration of data is so tedious that it overshadows the cause and analysis. It is necessary to briefly describe the results.

18.    Line 274-277: Please provide reference (references) for the process. And remove the "-" in line 277.

19.    Line 289: Change 140C to 140 ℃.

20.    The note at the bottom of the all the tables about the superscript letters needs to be corrected. They should not all be "a-c".

21.    "4. Conclusions": Why not mention the protein content of high-milk protein dispersions?

22.    The reference format is inconsistent. Please revise according to the requirements of the journal. Such as line 542, "A. Sereys et al. Potential…".

23.    This R&D section MUST be improved. Sufficient discussion SHOULD be elaborated and refined. Relationships between Tables and Figures MUST be studied, described and discussed in detail with references.

Author Response

  • Line 15: In the abstract, the full name of "MPC 85" should be given when the first time it is used. And "Three fresh lots of MPC 85 were obtained from a commercial manufacturer.", why is the detail of MPC included in the abstract?

AU: Removed from the abstract as suggested.  

  • Line 16: Change "SHMP at 0, 0.15, and 0.25% (w/w)" to "SHMP at 0%, 0.15%, and 0.25% (w/w)", and the same goes for line 17, line 23, etc.

AU: Corrected as suggested in the whole manuscript.

  • Be sparing with abbreviations in keywords and paper title: only abbreviations firmly established in the field may be eligible.

AU: Removed the abbreviations from the manuscript title and keywords. Keywords also updated.

  • Line 22: Change P>0.05 to P> 0.05. P should be italicized in the manuscript, and the same goes for line 28, line 175, etc.

AU: All fixed.

  • Line 29: I think the keyword "Beverage" is too broad, maybe you should use "high-protein beverage" or other specific word.

AU: Keywords updated

  • All the temperatures are written without spaces behind the numbers, such as line 68, 70, 73, etc.

A similar problem occurs in line 70 "7.00±0.05", so the author is asked to check the whole manuscript carefully.

AU: All fixed.

  • Line 76: Is this manufacturer's information for oil bath equipment?

AU: Yes, the (Narang Scientific Works Pvt. Ltd., New Delhi, India) is the manufacturer information.

  • Line 92: Please add instrument parameters and sampling details in "2.1.3".

AU; provided the reference.

  • Line 120: "as explained by [10]", please check the reference style and follow the guidelines of this journal.

Fixed.

  • Line 134: "L*, a*, and b*", "L, a and b" should be in italicized in the manuscript.

Fixed.

Line 154: What is the detail of P-NMR? And why are the details of phosphate and MPC included in this section? Is it more appropriate to put them in "2. Materials and Methods"?

The full name of NMR was mentioned as recommended “Phosphorus-31 nuclear magnetic resonance.” This section was removed under 2.1 section.

  • Line 168: Here it says "calcium ions", but in line 54 it says"Ca+2", so please write be consistent.

All fixed and used with using calcium ions in the manuscript.

  • Line 169-176: Why not describe the two SHMP levels (0.15 and 0.25%) together, since the trends are the same?

Modified as recommended

And throughout the whole tables, the absence of spaces before and after the "±".

All fixed.

  • In "3.1. Heat Coagulation Time (HCT)", is the reason for the increase in HCT caused by the SHMP chain length consistent at 6% and 8% protein content?

AU: Yes, both 6% and 8% protein levels behaved similar.

  • Line 200: Change "rates" to "rate".

Fixed

  • Table 2: What is the meaning of the "*" after the 8% "Heated", which is not stated in the Note?

AU: A note was added under the tables for the “*”. It means that no significant difference was observed between the samples and the control

  • In "3.2 Apparent Viscosity", the enumeration of data is so tedious that it overshadows the cause and analysis. It is necessary to briefly describe the results.

Edited as recommended.

  • Line 274-277: Please provide reference (references) for the process.

Reference 15 was added

And remove the "-" in line 277. Removed.

  • Line 289: Change 140C to 140 ℃.

Fixed

  • The note at the bottom of the all the tables about the superscript letters needs to be corrected. They should not all be "a-c".

All fixed.

  • "4. Conclusions": Why not mention the protein content of high-milk protein dispersions?

Improved as recommended.

  • The reference format is inconsistent. Please revise according to the requirements of the journal. Such as line 542, "A. Sereys et al. Potential…".

Fixed.

  • This R&D section MUST be improved. Sufficient discussion SHOULD be elaborated and refined. Relationships between Tables and Figures MUST be studied, described and discussed in detail with references.

Improved based on the prior comments

Reviewer 2 Report

Comments and Suggestions for Authors

The study evaluated the effect of different chain length of sodium hexametaphosphate on the physiochemical properties of heat treated high‐milk protein dispersions. The manuscript requires major improvements in terms of writing before being considered it for publication. It is very difficult to follow, the level of English language requires major improvement. In some cases, the authors are more focused on presenting the data without extracting the essential from them. 

Below, some specific comments: 

In the experimental design part, the authors should indicate the chain length of SHPM,  the volume of SHPM added in the MPC 85 solution. Indicate the necessary heating time to reach the required temperature and how was assessed. Please indicate what was the maximum capacity of the glass vials, the characteristics of the caps used to close the glass vials.

Based on the selected temperature, the authors performed the heat treatment under UHT conditions which is performed usually in thin layer. Therefore, how do authors know that every particle reached the desired temperature?

Line 83 - It is not clear why for heat coagulation time, it was chosen to heat treat only 2 ml of sample. What is the difference compared with what was mentioned at line 74?

Line 103 – why did the authors used front face fluorescence spectroscopy for liquid samples?

Line 114 – what represent 290/340 nm and 290/350 nm?

Lines 119 – 120 – what samples have been precipitated? After heat treatment?

Line 135 – what do the authors mean by 2 g of powder? I assumed that the samples were liquid.

Line 142- indicate what was used to dilute the sample.

Line 144 – indicate the material of the cuvette.

Lines 151 – 156 should be moved to the materials section.

Line 159 – explain the abbreviation.

Table 1 – explain the abbreviation in the legend of L, M, S. SHMP level is the concentration of SHPM. If yes, please change the word. For the statistical evaluation, indicate between which parameters have been performed the comparisons? Based on the protein content, based on the SHMP concentration, based on SHMP chain length? Same comment for the other tables.

Line 172 -173 – please reformulate “When 0.25% of Long, medium, and small SHMP had an HCT of 17.13, 18.46, and 19.29 min, respectively”

Lines 171 -174  can be combined to just one sentence.

Line 188 – p>0.05 indicates no significant effect. The authors declare the opposite.

Line 200 – indicate why did you choose to test the apparent viscosity at 100 s-1 shear rate.

Line 207 – p>0.05 indicates no significant effect. The authors declare the opposite. Check also line 228

Table 5 and table 6  - indicate in the legend the meaning of *

Comments on the Quality of English Language

The level of English language requires major improvements.

Author Response

The study evaluated the effect of different chain length of sodium hexametaphosphate on the physiochemical properties of heat treated high‐milk protein dispersions. The manuscript requires major improvements in terms of writing before being considered it for publication. It is very difficult to follow, the level of English language requires major improvement. In some cases, the authors are more focused on presenting the data without extracting the essential from them. 

Below, some specific comments: 

In the experimental design part, the authors should indicate the chain length of SHPM,  the volume of SHPM added in the MPC 85 solution. Indicate the necessary heating time to reach the required temperature and how was assessed. Please indicate what was the maximum capacity of the glass vials, the characteristics of the caps used to close the glass vials.

Based on the selected temperature, the authors performed the heat treatment under UHT conditions which is performed usually in thin layer. Therefore, how do authors know that every particle reached the desired temperature?

  • Line 83 - It is not clear why for heat coagulation time, it was chosen to heat treat only 2 ml of sample. What is the difference compared with what was mentioned at line 74?

In line 74, 3mL sample were used in heating (mimicking the UHT process) to get an enough sample to perform the rest of the analyses for the study. While in the heat coagulation test, 2 mL were enough for running the test as the aim of the analysis is to determine the time that needed for the sample to form coagulates (spoil).

  • Line 103 – why did the authors used front face fluorescence spectroscopy for liquid samples?

Liquid samples are so concentrated and opaque, so they can’t get tested using the right angle. Therefore, the front face was needed to detect the FAST index in the whole sample. Right angle was used on the transparent samples after they precipitated using the sodium acetate so we were able to detect the proteins in the serum phase.

  • Line 114 – what represent 290/340 nm and 290/350 nm?
    Tryptophan fluorescence spectra of unheated and heated samples were obtained at a wavelength of 290 nm for excitation and at 340 nm and 350 nm emission wavelengths, respectively. Updated in the manuscript.

  • Lines 119 – 120 – what samples have been precipitated? After heat treatment?
    In right angle method, all samples were precipitated using the sodium acetate to get the transparent sample where the proteins in the serum phase can get measured.

  • Line 135 – what do the authors mean by 2 g of powder? I assumed that the samples were liquid.

It was a typo, it was changed to About 2 mL of the sample… Line 143.

  • Line 142- indicate what was used to dilute the sample.

Added “diluted with distilled water”

  • Line 144 – indicate the material of the cuvette.

Added “plastic disposable cuvette”

  • Lines 151 – 156 should be moved to the materials section.

This section was removed under 2.1 section.

  • Line 159 – explain the abbreviation.

HPB is a high-protein dairy-based beverage. Got fixed.

  • Table 1 – explain the abbreviation in the legend of L, M, S. SHMP level is the concentration of SHPM. If yes, please change the word. For the statistical evaluation, indicate between which parameters have been performed the comparisons? Based on the protein content, based on the SHMP concentration, based on SHMP chain length? Same comment for the other tables.

L, M, and S were written as long, medium and short in all tables. The comparison was done within the column for the same protein content. A note was added in the footnote of the table to clarify this point.

  • Line 172 -173 – please reformulate “When 0.25% of Long, medium, and small SHMP had an HCT of 17.13, 18.46, and 19.29 min, respectively”
    Done as recommended.

  • Lines 171 -174  can be combined to just one sentence.

Edited as recommended

  • Line 188 – p>0.05 indicates no significant effect. The authors declare the opposite.

Fixed the p value. Yes, there was a significant difference (p <0.05)

  • Line 200 – indicate why did you choose to test the apparent viscosity at 100 s-1 shear rate.

That’s when the viscosity is more stabilized and has the least changes while continuing the measurement. Also, this is the shear rate used in the reference.

  • Line 207 – p>0.05 indicates no significant effect. The authors declare the opposite. Fixed

Check also line 228. Checked

  • Table 5 and table 6  - indicate in the legend the meaning of *

A note was added under the tables for the “*”. It means that no significant difference was observed between the samples and the control

Round 2

Reviewer 1 Report

Comments and Suggestions for Authors

The authors have made a  revision of their manuscript and have greatly improved its quality.

Author Response

No comments to respond. 

Reviewer 2 Report

Comments and Suggestions for Authors

The authors did not respond to all the comments that have been addressed by the reviewer.

Indicate the necessary heating time to reach the required temperature and how was assessed. Please indicate what was the maximum capacity of the glass vials, the characteristics of the caps used to close the glass vials. This is an important aspect that should be clarified in order to allow the reader to understand the real experimental conditions performed in the study.

The authors mentioned that they have used 3 mL of sample to mimic the UHT process. My opinion is that this statement is not correct. To mimic the UHT conditions the heat treatment is performed in glass capillaries.

 I asked the authors to indicate the volume of SHPM added in the MPC 85 solution and the authors did not respond to this question. Probably, the question was not clear formulated. They mentioned some concentrations, but it is not clear if this was the concentration of the SHPM before adding it to MPC, or the final concentration of SHPM in the MPC solution.

The authors should include the explanation given for using front face fluorescence spectroscopy also in the revised manuscript.

At the first evaluation the authors were asked to reformulate the sentence “When 0.25% of Long, medium, and small SHMP had an HCT of 17.13, 18.46, and 19.29 min, respectively. This statement was not reformulated in the revised version of the manuscript

In Table 5 and table 6  - the authors were asked to indicate in the legend the meaning of *

They replied that a note was added under the tables for the “*”. It means that no significant difference was observed between the samples and the control

The answer provided by the authors is not correct. If the authors did not find significant differences between samples, they should indicate in the column the same letter. By using the answer provided by the authors, the readers understands that the comparison was performed based on the control sample, which is not correct, as the test declared by the authors is Tukey. When the comparison is made based on the control, it is recommended to use Dunnet test.  The authors should clarify this aspect.

Comments on the Quality of English Language

Although some improvement have been recorded, there are many aspects where English language still requires major improvements. 

Author Response

  • Indicate the necessary heating time to reach the required temperature and how was assessed. Please indicate what was the maximum capacity of the glass vials, the characteristics of the caps used to close the glass vials. This is an important aspect that should be clarified in order to allow the reader to understand the real experimental conditions performed in the study. The authors mentioned that they have used 3 mL of sample to mimic the UHT process. My opinion is that this statement is not correct. To mimic the UHT conditions, heat treatment is performed on glass capillaries.

AU: Please see lines 79-88.

  • I asked the authors to indicate the volume of SHPM added in the MPC 85 solution and the authors did not respond to this question. Probably, the question was not clear formulated. They mentioned some concentrations, but it is not clear if this was the concentration of the SHPM before adding it to MPC, or the final concentration of SHPM in the MPC solution.

AU: Clarified in Lines 79-80

  • The authors should include the explanation given for using front face fluorescence spectroscopy also in the revised manuscript.

AU: Clarified in lines 114-117.

  • At the first evaluation the authors were asked to reformulate the sentence “When 0.25% of Long, medium, and small SHMP had an HCT of 17.13, 18.46, and 19.29 min, respectively. This statement was not reformulated in the revised version of the manuscript

Reformulated as recommended. Please see line 181-183.

  • In Table 5 and table 6  - the authors were asked to indicate in the legend the meaning of *. They replied that a note was added under the tables for the “*”. It means that no significant difference was observed between the samples and the control

The answer provided by the authors is not correct. If the authors did not find significant differences between samples, they should indicate in the column the same letter. By using the answer provided by the authors, the readers understand that the comparison was performed based on the control sample, which is not correct, as the test declared by the authors is Tukey. When the comparison is made based on the control, it is recommended to use Dunnet test.  The authors should clarify this aspect.

AU: You are correct. The comparison was done within the same column. The sentence in the table footnote was corrected. “*” Means that no significant difference was observed within the same column.

Round 3

Reviewer 2 Report

Comments and Suggestions for Authors

The authors have improved the content of the manuscript sufficiently to warrant the publication in Foods.